# Anti-Neuroinflammatory Effect of the Ethanolic Extract of Black Ginseng through TLR4-MyD88-Regulated Inhibition of NF-κB and MAPK Signaling Pathways in LPS-Induced BV2 Microglial Cells

**DOI:** 10.3390/ijms242015320

**Published:** 2023-10-18

**Authors:** Kwan-Woo Kim, Young-Seob Lee, Bo-Ram Choi, Dahye Yoon, Dae Young Lee

**Affiliations:** Department of Herbal Crop Research, National Institute of Horticultural and Herbal Sciences, Rural Development Administration, Eumseong 27709, Republic of Korea; swamp1@naver.com (K.-W.K.); youngseoblee@korea.kr (Y.-S.L.); bmcbr@korea.kr (B.-R.C.); dahyeyoon@korea.kr (D.Y.)

**Keywords:** black ginseng, anti-neuroinflammation, BV2-microglial cells, NF-κB, MAPK, TLR4, MyD88

## Abstract

Korean ginseng (*Panax ginseng*) contains various ginsenosides as active ingredients, and they show diverse biological activities. Black ginseng is manufactured by repeated steaming and drying of white ginseng, which alters the polarity of ginsenosides and improves biological activities. The aim of the present investigation was to examine the anti-neuroinflammatory effects of the ethanolic extract of black ginseng (BGE) in lipopolysaccharide (LPS)-induced BV2 microglial cells. Pre-treatment with BGE inhibited the overproduction of pro-inflammatory mediators including nitric oxide (NO), prostaglandin E2 (PGE_2_), inducible nitric oxide synthase (iNOS), cyclooxygenase-2 (COX-2), interleukin-6 (IL-6), and tumor necrosis factor-α (TNF-α) in LPS-induced BV2 cells. In addition, BGE reduced the activation of nuclear factor kappa-light-chain-enhancer of activated B cells (NF-κB), p38 mitogen-activated protein kinase (MAPK), and c-jun N-terminal kinase (JNK) MAPK signaling pathways induced by LPS. These anti-neuroinflammatory effects were mediated through the negative regulation of the toll-like receptor 4 (TLR4)/myeloid differentiation primary response 88 (MyD88) signaling pathway. Among the four ginsenosides contained in BGE, ginsenosides Rd and Rg3 inhibited the production of inflammatory mediators. Taken together, this investigation suggests that BGE represents potential anti-neuroinflammatory candidates for the prevention and treatment of neurodegenerative diseases.

## 1. Introduction

Microglial cells are considered macrophage-type cells in the central nervous system (CNS), and they have an important role in homeostasis and a major type of active immune defense in the CNS. Microglial cells comprise 10–15% of total cells in the brain, and the activation of microglial cells is closely related to neuroinflammation [1]. The cells maintain homeostasis in the CNS, especially the brain, and when they detect pathological changes such as invasion of external pathogens and damage to nerve cells, they are quickly activated to produce various pro-inflammatory mediators and cytokines, including nitric oxide (NO), prostaglandin E2 (PGE_2_), inducible nitric oxide synthase (iNOS), cyclooxygenase-2 (COX-2), and pro-inflammatory cytokines, such as interleukin-1β (IL-1β), IL-6, and tumor necrosis factor-α (TNF-α) [2,3]. However, the over-production of pro-inflammatory mediators could cause oxidative damage to the lipid components of cells, proteins, and DNA, resulting in the death of nerve cells [4]. In addition, a continuous inflammatory response in the CNS can cause neuronal injuries, leading to the development of neurodegenerative diseases including Alzheimer’s disease (AD), Parkinson’s disease (PD), multiple sclerosis, and acquired immune deficiency syndrome (AIDS) dementia [5]. Therefore, one of the treatments for neurodegenerative diseases is to effectively regulate the abnormal secretion of inflammatory mediators and cytokines, which play a leading role in the progression of neurodegenerative diseases.

Korean ginseng (*Panax ginseng* C. A. Meyer) is well known as a valuable medicinal plant belonging to the Panax genus of Araliaceae, and it is reported to have various ginsenosides that show diverse biological activities. Fresh ginseng could be processed into Taegeuk ginseng, black ginseng, and red ginseng depending on how fresh ginseng is steamed and dried, which improves storage and efficacy compared to fresh ginseng [6]. Black ginseng is manufactured by steaming and drying white ginseng several times. The steaming process lowers the polarity of ginsenosides in white ginseng [7] and improves its biological activities compared to white or red ginseng, such as anti-obesity, anti-diabetes, anti-cancer, anti-inflammatory, anti-nociceptive, anti-oxidant, anti-hyperglycemic, and immune-modulating activities [8,9]. Recently, we reported that black ginseng strongly reduced the secretion of virulence factors including Hla, staphylococcal enterotoxin A (SEA), and staphylococcal exterotoxin B (SEB) in methicillin-resistant *Staphylococcus aureus* (MRSA) [10]. Although it has been reported that black ginseng-enriched chongmyeongtang extract not only improved memory impairment but also inhibited scopolamine or lipopolysaccharide (LPS)-induced neuroinflammation [11], the effect of black ginseng extract alone has not been reported. Therefore, as a part of our ongoing search for novel materials for the development of new health functional foods, this study describes the anti-neuroinflammatory effect of the ethanolic extract of black ginseng (BGE).

## 2. Results

### 2.1. The Effect of BGE on the Viability of BV2 Microglial Cells

The 3-(4,5-dimethylthiazol-2-yl)-2,5-diphenyl-2H-tetrazolium bromide (MTT) assay was conducted for the evaluation of the cytotoxicity of BGE in BV2 cells. The result of the MTT assay showed that BGE at 12.5~200 μg/mL concentration had no effect on BV2 cell viability (Figure 1).

### 2.2. The Inhibitory Effect of BGE on the Production of NO and PGE_2_ in LPS-induced BV2 Microglial Cells

It is being investigated whether BGE affects the LPS-induced production of NO and PGE_2_ in BV2 microglial cells. As shown in Figure 2, the production of NO and PGE_2_ increased upon stimulation with LPS. However, pre-treatment with BGE significantly inhibited both levels of NO and PGE_2_ in a dose-dependent manner.

### 2.3. The Inhibitory effect of BGE on the Expression of iNOS and COX-2 Proteins in LPS-induced BV2 Microglial Cells

The inhibition of the expression of inducible enzymes, including iNOS and COX-2, which are responsible for the production of NO and PGE_2_, respectively, could be an important target to prevent neuroinflammation. Therefore, we examined whether BGE inhibits LPS-induced overexpression of iNOS and COX-2 proteins in BV2 cells. As shown in Figure 3, we found that LPS markedly augmented iNOS and COX-2 protein levels. However, as a result of pre-treatment with BGE, especially significantly compared to the LPS group at 200 μg/mL concentration.

### 2.4. The Inhibitory Effect of BGE on the Production of Pro-Inflammatory Cytokines and the Expression of Those mRNA in LPS-Induced BV2 Microglial Cells

The activated microglia could induce excessive production of pro-inflammatory cytokines. Accordingly, we investigated whether BGE affected the LPS-induced production and mRNA expression of pro-inflammatory cytokines, including IL-6 and TNF-α, using enzyme-linked immunosorbent assay (ELISA) and quantitative real-time reverse transcription polymerase chain reaction (qRT-PCR) analysis. The production of IL-6 and TNF-α and their corresponding mRNA expression increased in response to LPS. However, pre-treatment with BGE attenuated both production (Figure 4A,B) and mRNA expression (Figure 4C,D) of those cytokines in LPS-induced BV2 microglial cells.

### 2.5. The Inhibitory Effect of BGE on the Activation of NF-κB Signaling Pathway in LPS-induced BV2 Microglial Cells

We next evaluated the regulatory activity of BGE on the activation of the nuclear factor kappa-light-chain-enhancer of activated B cells (NF-κB) signaling pathway. The phosphorylation and degradation of inhibitor kappa B (IκB)-α were attenuated by pre-treatment with BGE compared to those in the cells treated with LPS only (Figure 5A–C). In addition, the levels of the p65 subunit decreased in the nuclear fraction with BGE treatment (Figure 5D,E).

### 2.6. The Inhibitory Effect of BGE on the Activation of MAPK Signaling Pathway in LPS-induced BV2 Microglial Cells

Along with the activation of the NF-κB signaling pathway, we considered the inhibitory effect of BGE on the activation of mitogen-activated protein kinase (MAPK) signaling pathways. As a result, BGE was shown to inhibit the phosphorylation of p38 (Figure 6A) and c-jun N-terminal kinase (JNK) MAPKs (Figure 6C) in LPS-stimulated BV2 cells, while the phosphorylation of extracellular signal-regulated kinase (ERK) MAPK was not (Figure 6B).

### 2.7. The Inhibitory Effect of BGE on the Activation of TLR4/MyD88 Signaling Pathway in LPS-Induced BV2 Microglial Cells

To further elucidate the mechanism involved in the anti-neuroinflammatory effect of BGE, we evaluated the effect of BGE on mRNA levels of toll-like receptor 4 (TLR4) and myeloid differentiation primary response 88 (MyD88) in LPS-induced BV2 microglial cells. The mRNA levels of TLR4 in BV2 cells stimulated with LPS for 6 h significantly increased, but pre-treatment with BGE inhibited TLR4 mRNA expression (Figure 7A). Under the same condition, BGE suppressed the expression of MyD88 mRNA in a concentration-dependent manner in LPS-induced BV2 cells (Figure 7B).

### 2.8. Quantitative Analysis of Black Ginseng Extract by HPLC

Quantitative analysis of BGE was performed by HPLC. The contents of ginsenoside Rd, Rg3, Rk1, and Rg5 in BGE were calculated from the peak areas by interpolation to standard calibration curves (Table 1, Figure 8). BGE was found to contain ginsenoside Rd at 0.190 ± 0.011 mg/g, Rg3 at 2.060 ± 0.090 mg/g, Rk1 at 2.332 ± 0.085, and Rg5 at 2.294 ± 0.079 mg/g, respectively (Table 2).

### 2.9. The Inhibitory Effect of Ginsenosides in BGE on the Production of NO, IL-6, and TNF-α in LPS-Induced BV2 Microglial Cells

To determine which component causes the anti-neuroinflammatory effect of BGE, the effect of ginsenosides Rd, Rg3, Rk1, and Rg5 on the production of NO, IL-6, and TNF-α in LPS-induced BV2 microglial cells was conducted. Firstly, the MTT assay was conducted for the evaluation of the cytotoxicity of four ginsenosides in BV2 cells. The result of the MTT assay showed that ginsenoside Rd and Rg3 did not show cytotoxicity up to 80 μM concentrations, but ginsenoside Rk1 and Rg5 showed toxicity at 40 and 80 μM concentrations (Figure 9A). Afterwards, the effect on the production of NO, IL-6, and TNF-α induced by LPS in BV2 cells pre-treated with four ginsenosides within a non-toxic concentration range was confirmed. As shown in Figure 9B, treatment with three ginsenosides except for ginsenoside Rk1 significantly downregulated the production of NO in LPS-stimulated BV2 microglial cells, of which ginsenoside Rd had the greatest effect by 90.0%. In addition, ginsenoside Rd and Rg3 markedly inhibited the production of IL-6 and TNF-α in LPS-induced BV2 microglial cells, with the greatest effect of Rd on both IL-6 and TNF-α by 90.0% and 93.6%, respectively (Figure 9C,D).

## 3. Discussion

This investigation demonstrated that BGE appeared to have an anti-neuroinflammatory effect in LPS-induced BV2 microglial cells. The anti-neuroinflammatory activity of BGE was related to inhibition of NF-κB, p38 MAPK, and JNK MAPK signaling pathways via inactivation of the TLR4/MyD88 pathway. These inhibitory effects of BGE are thought to be mainly caused by the ginsenosides Rd and Rg3 contained in BGE.

The production of NO and PGE_2_ is a major biomarker for the activation of microglial cells, which is regulated by the enzyme activity of iNOS and COX-2, respectively. NO has been demonstrated to be a versatile molecule that plays important roles in the regulation of vascular tone and neurotransmission as well as the pathogenesis of acute and chronic inflammation. It is generated from various immune cells, including macrophages, microglia, monocytes, and other cells involved in immune reactions [12]. The overproduction of NO could lead to the occurrence of inflammatory diseases such as inflammatory bowel diseases, neurodegenerative diseases, and cancer [13,14,15]. NO is catalyzed by the enzymatic activity of three nitric oxide synthases (NOS), including endothelial NOS (eNOS), neuronal NOS (nNOS), and iNOS, and they transform L-arginine to NO and L-citrulline [16]. PGE_2_ is also one of the most typical lipid mediators that is significantly involved in the development of many inflammatory diseases. It is biosynthesized from arachidonic acid (AA) by the enzymatic effects of cyclooxygenases (COX) and PGE synthases (PGES) [17]. Among the various prostanoids generated from AA, PGE_2_ is the most abundant, and they are combined with four PGE receptors called E-type prostanoid receptors 1–4 (EP1-4) to exert a variety of physiological and pathological activities [18]. COX could be classified into two isoforms, COX-1 and COX-2. COX-1 is constitutively expressed to regulate numerous homeostatic functions, including platelet aggregation, hemostasis, and protection of the gastric mucosa. On the other hand, COX-2 is inducible, and various stimuli, including LPS, ILs, TNF, epidermal growth factor (EGF), and platelet activating factor (PAF), which are mainly associated with inflammation, could activate the expression of COX-2 [19]. Therefore, it is important to regulate the major inflammatory mediators, including NO, PGE_2_, iNOS, and COX-2. In this investigation, the pre-treatment with BGE inhibited the overproduction of NO and PGE_2_ and the overexpression of their responsible enzymes, iNOS and COX-2 proteins, in LPS-induced BV2 microglial cells (Figure 2 and Figure 3).

Pro-inflammatory cytokines are small secreted proteins from a broad range of various immune cells, like macrophages, lymphocytes, mast cells, and endothelial cells, and have an important role in acute and chronic inflammation through a complex and contradictory network of interactions between cells [20,21]. Cytokine is a general name, and it could be classified into various types depending on which cell it comes from and how it works: lymphokine made by lymphocytes, monokine from monocytes, chemokine made by chemotactic activities, interleukin from one leukocyte acting on other leukocytes, growth factors including platelet-derived growth factor (PDGF) and transforming growth factor (TGF), and TNF families [20]. Cytokines play various roles in normal CNS system function, including the regulation of sleep, neuroendocrine functions, neuronal development, normal aging, and inflammatory states from bacterial and viral infections of either the brain or the periphery [22]. In particular, excessive inflammatory cytokines caused by excessive inflammatory responses in the CNS could lead to the occurrence of various neurodegenerative diseases [22]. In this investigation, treatment with BGE resulted in decreased production and mRNA expression of IL-6 and TNF-α in LPS-induced BV2 microglial cells (Figure 4).

NF-κB is one of the key transcription factors for the induction of inflammatory-related gene expression [23]. Various stimuli, including pro-inflammatory cytokines, interferons, or LPS, can activate the NF-κB signaling pathway, inducing the phosphorylation and degradation of IκB-α in the cytosol and the translocation of p65 and p50 subunits into the nucleus [24]. It resulted in the binding of NF-κB subunits to DNA, leading to the activation of inflammatory mediators [25]. Therefore, the inactivation of NF-κB is regarded as a major target for the treatment of neuroinflammation. In this study, LPS-induced activation of the NF-κB signaling pathway was attenuated by pre-treatment with BGE by reducing the phosphorylation and degradation of IκB-α and the translocation into the nucleus of the p65 subunit in BV2 microglial cells (Figure 5).

MAPKs are the serine/threonine protein kinases family and contribute to the regulation of various cellular processes, including mitosis, metabolism, apoptosis, gene induction, cell survival, differentiation, and proliferation [26,27]. In mammals, MAPKs are classified as conventional MAPKs comprising ERK1/2, JNK1/2/3, p38 isoforms, and ERK5, and atypical MAPKs including ERK3/4, ERK7, and Nemo-like kinase (NLK) [28]. They are also important intracellular signaling pathways related to inflammatory responses [29], and various studies have reported that natural products inhibit inflammatory reactions by inactivating MAPK signaling pathways [30,31]. Our results showed that BGE inhibited LPS-induced activation of p38 and JNK MAPKs by inhibiting the phosphorylation of them in BV2 microglial cells (Figure 6).

TLRs are a large family of pattern recognition receptors (PRRs), which recognize pathogen- and damaged-associated molecular patterns (PAMPs/DAMPs) and play an important role in the initiation of innate immunity or the occurrence of inflammation through the sensing of body pathogens [32,33]. TLRs are classified into 13 types according to the interacting PAMP or DAMP, of which TLR4 is a transmembrane protein present in the cell membrane and mainly interacts with LPS [34]. The activation of TLR4 leads to the recruitment of specific adaptor molecules, which belong to the toll/interleukin-1 receptor (TIR) family, including MyD88 and TIR-domain-containing adapter-inducing interferon-β (TRIF), and affects the NF-κB and MAPK signaling pathways, consequently regulating the gene expression of inflammatory mediators [35,36]. Although TLR4 exhibits protective activity as an essential mediator of immune responses, inappropriate TLR responses could induce acute and chronic inflammation [37]. Our data revealed that BGE suppressed the mRNA expression of TLR4 and MyD88 in LPS-induced BV2 microglial cells in a dose-dependent manner (Figure 7), suggesting that BGE could exhibit an anti-neuroinflammatory effect through inactivation of TLR4/MyD88 signaling in LPS-stimulated microglial cells.

In the previous investigation, it was found that the main ingredients contained in BGE are ginsenoside Rd, Rk1, Rg5, and Rg3 [38], which could prevent the development of neurodegenerative diseases such as Alzheimer’s diseases, Parkinson’s diseases, or depression by inhibiting inflammatory responses in brain cells [39,40,41,42]. Our HPLC analysis results confirmed that BGE contains active ingredients including ginsenoside Rd, Rg3, Rk1, and Rg5, of which Rk1 is the most abundant and Rd is the least (Table 2 and Figure 8). The effect on the production of NO and pro-inflammatory cytokines induced by LPS of Rd was significantly higher than that of the other ginsenosides (Rg3, Rg5, Rk1) (Figure 9). The chemical structures of Rg3, Rg5, Rk1, and Rd are similar, but they have different sugar groups. In particular, Rd has one more glucose group at C-21 than other ginsenosides. This result indicates that the glucose group at carbon # 21 of Rd is one of the key factors in the anti-inflammatory effects of this molecule.

Based on these results, the anti-neuroinflammatory effect of BGE appears to be mainly caused by the biological action of the ginsenosides Rd and Rg3. According to the results of HPLC analysis, it was confirmed that BGE also contains other ingredients in addition to four ginsenosides, and it is considered necessary to investigate what they are and whether they exhibit anti-neuroinflammatory activity through further research.

## 4. Materials and Methods

### 4.1. The Preparation of Ethanolic Extract of Black Ginseng

The ethanolic extract of black ginseng (BGE) was prepared based on a previously reported investigation [43].

### 4.2. Chemicals and Reagents

RPMI1640 media, fetal bovine serum (FBS), penicillin-streptomycin, phosphate-buffered saline (PBS), and trypsin-EDTA (TE) were purchased from Gibco BRL Co. (Grand Island, NY, USA). Lipopolysaccharide (LPS, O55:B5), dimethyl sulfoxide (DMSO), 3-(4,5-dimethylathiazol-2yl)-2,5-diphenyltetrazoleum (MTT), ammonium persulfate (APS), and tween-20 were obtained from Merck (Darmstadt, Germany). IL-6 and TNF-α ELISA kit were purchased from R&D System (Minneapolis, MN, USA). RIPA lysis buffer, protease and phosphatase inhibitor cocktail, and NE-PER™ Nuclear and Cytoplasmic Extraction Reagents were purchased from Thermo Fisher Scientific (Waltham, MA, USA). Nitrocellulose (NC) membrane, bis-acrylamide solution, Tris-HCl, and tetramethylethylenediamine (TEMED) were obtained from Bio-Rad Laboratories, Inc. (Hercules, CA, USA). Sodium dodecyl sulfate (SDS), Tris-Glycine SDS Buffer, Tris-Glycine Native Buffer, and Tris-buffered saline (TBS) were purchased from GenDepot, and skimmed milk powder was obtained from BD Biosciences (Franklin Lakes, NJ, USA). The primary antibodies, including anti-iNOS, anti-COX-2, anti-IκB-α, anti-p-IκB-α, anti-p65, anti-p-ERK, anti-ERK, anti-p-JNK, anti-JNK, anti-p-p38, and anti-p38, were obtained from Cell Signaling Technology Inc. (Danvers, MA, USA), and anti-β-actin primary antibodies and anti-PCNA were obtained from Santa Cruz Biotechnology Inc. (Dallas, TX, USA). Secondary antibodies, including anti-mouse, anti-rabbit, and anti-goat, are from Merck Millipore Co. (Darmstadt, Germany). Ginsenosides, including ginsenoside Rg3, Rk1, and Rg5, were obtained from Sigma–Aldrich, and ginsenoside Rd was purchased from Chemfaces (Wuhan, Hubei, China).

### 4.3. Cell Culture and Sample Preparation

BV2 cells were maintained at 5 × 10^5^ cells/mL in dishes of 100 mm in diameter in RPMI1640 supplemented with 10% (*v*/*v*) heat-inactivated FBS, penicillin G (100 units/mL), streptomycin (100 μg/mL), and L-glutamine (2 mM). They were cultured at 37 °C in a humidified atmosphere containing 5% CO_2_. BGE and ginsenosides were dissolved using DMSO, which were diluted by concentration using RPMI1640 media to be treated on cells.

### 4.4. MTT Assay for Cell Viability

BV2 cells were cultured and treated with BGE and four ginsenosides at different concentrations for 24 h, and an MTT assay was performed to determine cell viability based on the previously reported protocol [14]. The absorbance was measured at 540 nm wavelength using a Multiskan Microplate Reader (Thermo Fisher). The optical density of the formazan solution from the control group (untreated group) was considered to indicate 100% viability.

### 4.5. Determination of Nitrite (NO Production)

Cells were cultured and treated with BGE, and four ginsenosides followed LPS stimulation (1 μg/mL) for 24 h, and the level of NO was estimated using the Griess reaction described in the previous report [44]. The nitrite concentration was measured using a Multiskan Microplate Reader (Thermo Fisher) with a 540 nm wavelength.

### 4.6. Western Blot Analysis

The detailed procedures for Western blot analysis were referenced in a previous investigation [44]. Cells were pre-treated with BGE for 3 h and stimulated with LPS (1 μg/mL) for 24 h (iNOS, COX-2) or 1 h (NF-κB and MAPK). β-Actin and PCNA were used as internal controls.

### 4.7. Assays for PGE_2_, IL-6, and TNF-α

The concentration of PGE_2_, IL-6, and TNF-α contained in culture media was determined using profit ELISA kits. The assays were performed according to the manufacturer’s instructions, and three independent replicates were performed.

### 4.8. Preparation of Cytosolic and Nucelar Fractions

Nuclear and cytosolic extracts of the cells were obtained using the NE-PER™ nuclear and cytoplasmic extraction reagents. Each extract was subsequently lysed in accordance with the instructions of the manufacturer.

### 4.9. Quantitative Real-Time Reverse Transcription PCR (qRT-PCR)

Total RNA was isolated from the cells using Trizol (Invitrogen, Carlsbad, CA, USA), according to the manufacturer’s recommendations, and spectrophotometrically quantified (at 260 nm wavelength). cDNA was obtained using isolated total RNA (4 μg) using the high-capacity RNA-to-cDNA kit (Applied Biosystems, Carlsbad, CA, USA). Relative gene expression was quantified using qTOWER3 G Real-Time PCR Thermal Cycle (Analytik Jena, Jena, Germany) with SYBR Green PCR Master Mix (Applied Biosystems, Carlsbad, CA, USA) and oligonucleotide primers (Table 3) designed by CosmoGenetech (Seoul, Republic of Korea). The cycle number at the linear amplification threshold (C_t_) values of genes was normalized to the C_t_ values of the endogenous control gene (GAPDH) calculated with the comparative Ct method (2^–ΔΔCt^) using the qPCR software (software version 4.0) developed by Analytik Jena (Jena, Germany). The analysis was conducted three times, independently.

### 4.10. HPLC Analytical Conditions

Chromatographic analysis was performed using an Agilent 1260 Infinity Ⅱ system with a diode array detector. The analysis was conducted using a Zorbax Eclipse Plus C18 column (150 × 4.6 mm, 3.5 μm) with an injection volume of 10 μL at 30 °C. The mobile phase was composed of water (A) and acetonitrile (B), in the following gradient system: 0–10 min, 18–10% B; 10–30 min, 27% B; 30–40 min, 27–30% B; 40–55 min, 30–51% B; 55–66 min, 51–70% B; 66–71 min, 70–95% B; 71–76 min, 95% B; 76–77 min, 95–18% B; and 77–88 min, 18% B. The flow rate for HPLC analysis was set to 1.6 mL/min, while the detection wavelength was set to 203 nm. Each sample was analyzed three times. Standard samples, including ginsenoside Rd, Rg3, Rk1, and Rg5, were dissolved in methanol. BGE was dissolved in 10% acetonitrile at a concentration of 10 mg/mL and then filtrated using a 0.45 μm PVDF filter.

### 4.11. Statistical Analysis

The data are the mean ± standard deviation (SD) of at least three independent experiments. One-way analysis of variance (one-way ANOVA) and subsequent Tukey’s multiple comparison tests were employed to compare three or more groups. All the statistical analyses were performed using GraphPad Prism software (version 3.03, GraphPad Software Inc., La Jolla, CA, USA).

## 5. Conclusions

In summary, our study demonstrated that the ethanolic extract of black ginseng was shown to have inhibitory effects toward overproduction of pro-inflammatory mediators including NO, PGE_2_, IL-6, and TNF-α and overexpression of iNOS and COX-2 proteins in LPS-stimulated BV2 microglial cells. In addition, BGE reduced the activation of NF-κB, p38 MAPK, and JNK MAPK signaling pathways, and these phenomena were confirmed to be regulated by the inactivation of the TLR4/MyD88 pathway. Through the analysis of the components of BGE, the contents of the active ingredients of BGE, including ginsenoside Rd, Rg3, Rk1, and Rg5, of which ginsenoside Rd and Rg3 inhibited the production of NO, IL-6, and TNF-α. Taken together, our results suggest that BGE represents significant anti-neuroinflammatory effects, showing the availability of black ginseng as a potential candidate for the development of treatments for neurodegenerative diseases and as a raw material health functional foods to improve cognitive and memory deficiencies.

## Figures and Tables

**Figure 1 ijms-24-15320-f001:**
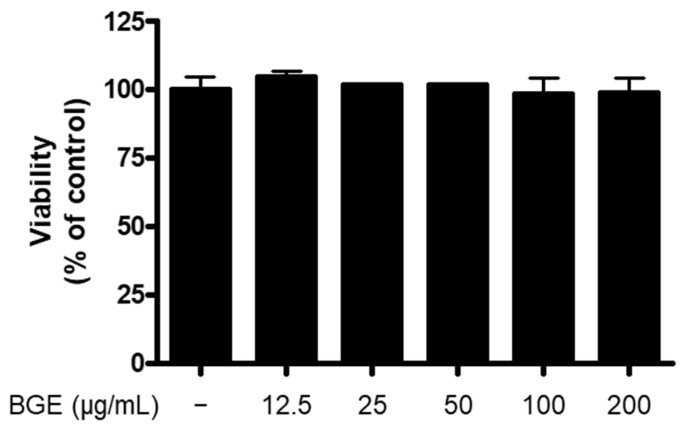
The effect of BGE on BV2 cell viability. Cells were treated with indicated concentrations of BGE for 24 h. The viability was determined by MTT assay.

**Figure 2 ijms-24-15320-f002:**
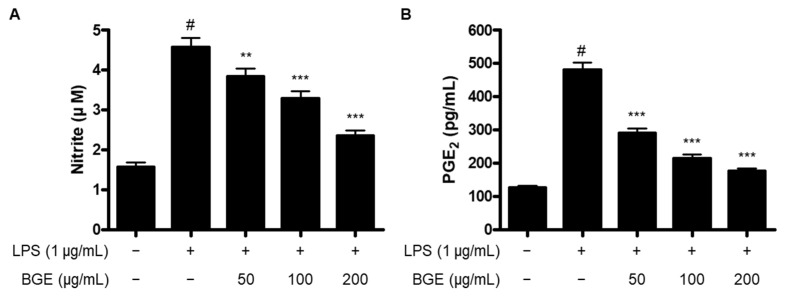
The effect of BGE on the production of NO (**A**) and PGE_2_ (**B**) in LPS-induced BV2 microglial cells. Data represent the mean values of three independent experiments ± SD. # *p* < 0.001 compared to the control group; ** *p* < 0.01, and *** *p* < 0.001 compared to the LPS-treated group.

**Figure 3 ijms-24-15320-f003:**
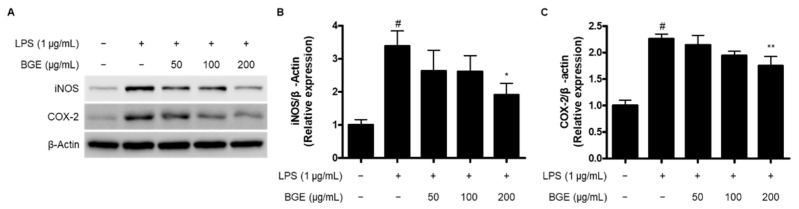
The effect of BGE on the expression of iNOS and COX-2 proteins in LPS-induced BV2 microglial cells. (**A**) Representative blots from three independent experiments are shown. (**B**,**C**) Relative expression of iNOS and COX-2. # *p* < 0.001 compared to the control group; * *p* < 0.05, and ** *p* < 0.01 compared to the LPS-treated group.

**Figure 4 ijms-24-15320-f004:**
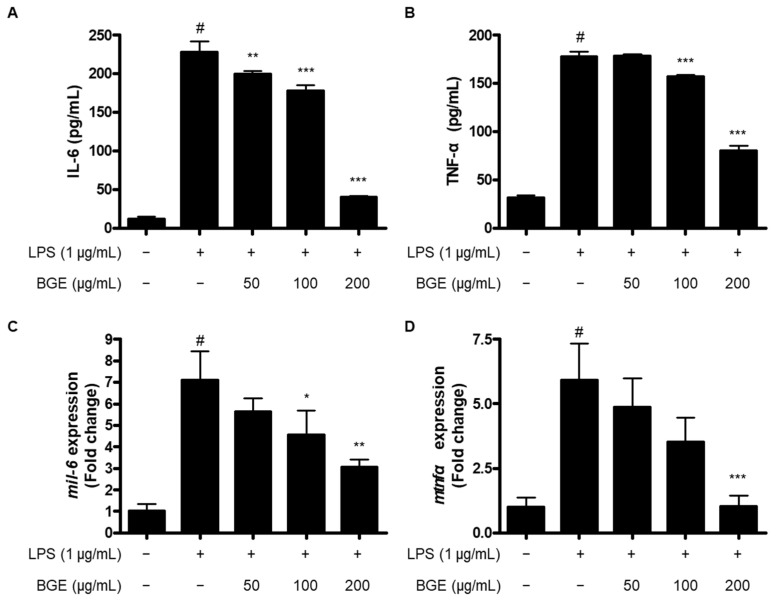
The effect of BGE on IL-6 (**A**,**C**) and TNF-α (**B**,**D**) in LPS-induced BV2 microglial cells. Data represent the mean values of three independent experiments ± SD. # *p* < 0.001 compared to the control group; * *p* < 0.05, ** *p* < 0.01, and *** *p* < 0.001 compared to the LPS-treated group.

**Figure 5 ijms-24-15320-f005:**
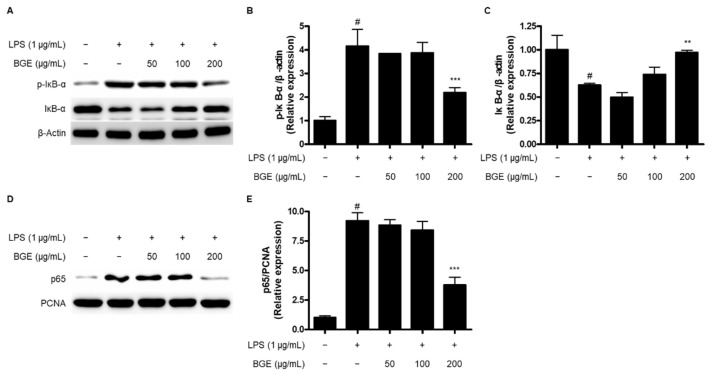
The effect of BGE on the LPS-induced activation of NF-κB signaling pathway in BV2 microglial cells. Data represent the mean values of three independent experiments ± SD. (**A**) Representative blots for the expression of p-IκB-α and IκB-α. (**B**,**C**) Relative expression of p-IκB-α and IκB-α. (**D**) Representative blots for the expression of p65. (**E**) Relative expression of p65. # *p* < 0.001 compared to the control group; ** *p* < 0.01, and *** *p* < 0.001 compared to the LPS-treated group.

**Figure 6 ijms-24-15320-f006:**
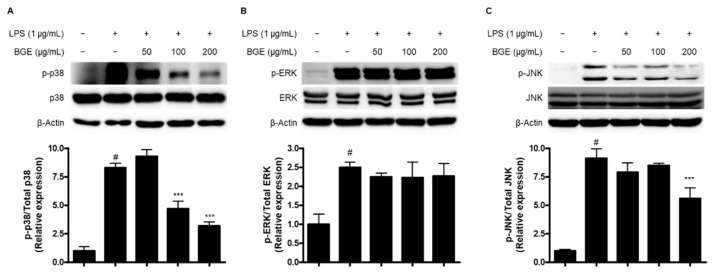
The effect of BGE on the LPS-induced activation of (**A**) p38, (**B**) ERK, and (**C**) JNKMAPK signaling pathway in BV2 microglial cells. Data represent the mean values of three independent experiments ± SD. # *p* < 0.001 compared to the control group; *** *p* < 0.001 compared to the LPS-treated group.

**Figure 7 ijms-24-15320-f007:**
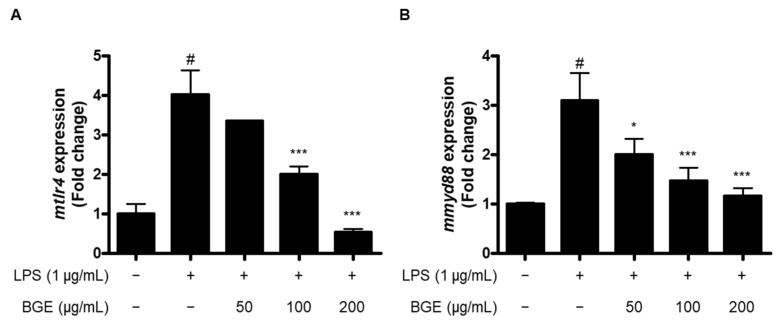
The effect of BGE on the LPS-induced mRNA expression of (**A**) TLR4 and (**B**) MyD88 in BV2 microglial cells. Data represent the mean values of three independent experiments ± SD. # *p* < 0.001 compared to the control group; * *p* < 0.05; and *** *p* < 0.001 compared to the LPS-treated group.

**Figure 8 ijms-24-15320-f008:**
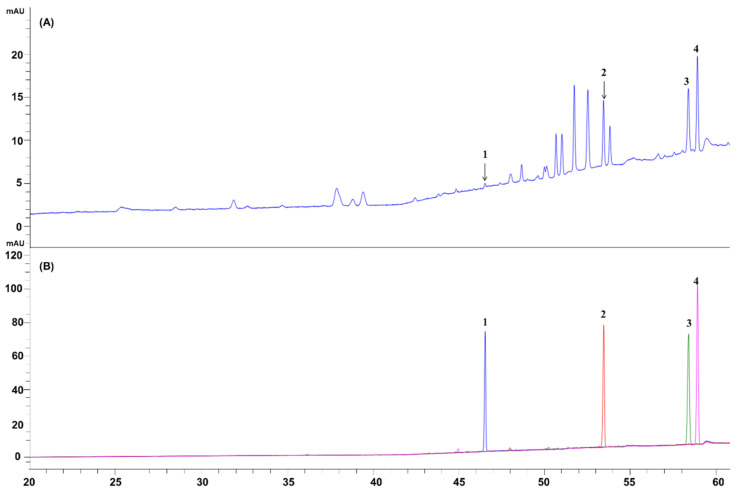
HPLC chromatogram of (**A**) BGE and (**B**) overlay chromatogram of ginsenosides [ginsenoside Rd (1), Rg3 (2), Rk1 (3), and Rg5 (4)].

**Figure 9 ijms-24-15320-f009:**
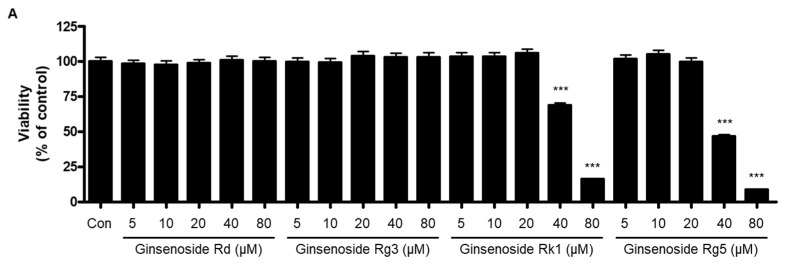
The effect of ginsenosides in BGE on BV2 cell viability (**A**) and on the LPS-induced production of NO (**B**), IL-6 (**C**), and TNF-α (**D**) in BV2 microglial cells. (**A**) The effect of ginsenosides in BGE on BV2 cell viability. (**B**) The concentration of nitrite was determined by Griess assay. (**C**,**D**) The concentrations of IL-6 and TNF-α were determined by ELISA. All data represent the mean values of three independent experiments ± SD. # *p* < 0.001 compared to the control group; * *p* < 0.05, and *** *p* < 0.001 compared to the LPS-treated group.

**Table 1 ijms-24-15320-t001:** Parameters for quantitative analysis of ginsenosides by HPLC.

Compounds	Retention Time (min)	Calibration Equation ^a^	Correlation Coefficient (r^2^)	Linear Range (μg/mL)
Ginsenoside Rd	46.567	y= 1.8886x+ 1.0527	0.9991	1.56~200
Ginsenoside Rg3	53.452	y= 2.2189x+ 1.8239	0.9958	1.56~200
Ginsenoside Rk1	58.347	y= 2.4681x+ 0.5814	0.9983	1.56~200
Ginsenoside Rg5	58.845	y= 3.2204x+ 2.9636	0.9979	1.56~200

^a^ y: peak area, x: concentration of standards (mg/mL).

**Table 2 ijms-24-15320-t002:** Quantitative Analysis of ginsenoside content in BGE.

Compounds	Contents (mg/g)	RSD * (%)
Ginsenoside Rd	0.190 ± 0.011	5.768
Ginsenoside Rg3	2.060 ± 0.090	4.373
Ginsenoside Rk1	2.332 ± 0.085	3.631
Ginsenoside Rg5	2.294 ± 0.079	3.448

Values are the mean ± SEM. * RSD: Relative Standard Deviation.

**Table 3 ijms-24-15320-t003:** Primer sequences used in this investigation.

Gene Name	Forward Primers (5′-3′)	Reverse Primers (5′-3′)
*IL-6*	ACTTCACAAGTCGGAGGCTT	TGTTGCTACTATCGTGAACGT
*TNF-α*	CCAGACCCTCACACTCACAA	CGGCTACCCAACATGGAACA
*TLR4*	GGCAACTTGGACCTGAGGAG	CCATGT GTCCATGGGCTCT
*MyD88*	ACTGAAGGAGCTGAAGTCGC	CGAAAAGTTCCGGCGTTTGT
*GAPDH*	TTCACCACCATGGAGAAGGC	AGTACTGGTGTCAGGTACGG

## Data Availability

The data presented in this study are available on request from the corresponding authors.

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
