# Peer review of "Anti-Neuroinflammatory Effect of the Ethanolic Extract of Black Ginseng through TLR4-MyD88-Regulated Inhibition of NF-κB and MAPK Signaling Pathways in LPS-Induced BV2 Microglial Cells"

_ijms, 2023, doi:10.3390/ijms242015320_

Round 1
Reviewer 1 Report
The topic of the work is not very original, but it is a continuation or rather an extension of research on the medicinal properties of black ginseng extracts by some of the authors (probably from the same, larger research team), the results of which were previously published (Manuscript ID ijms- 2520355). Each new approach to a specific topic allows you to expand your knowledge in a specific area. Sometimes it confirms and sometimes excludes available data. In this aspect, the results presented by the authors are worth publishing. For the methods, I have previously reviewed other works by these authors. I then submitted comments and now I see that the authors also took them into account in the current manuscript, which helped avoid repeating random mistakes. I have no comments on the consistency of the applications. In my opinion, they refer to the assumed goal and the results of the research conducted. Similarly, the discussion is coherent and logical. References are selected and cited correctly. Formatting appropriate to IJMS requirements has also been preserved.
The English language is basically correct.
Author Response
Dear Editors and Reviewers,
We greatly appreciate your kind consideration on our manuscript. According to the Reviewers’ comments, some parts of the manuscript have been corrected and revised.
Editor comments: Please rewrite lines 81-86, 130-133, 147-151, 167-168, 320-325. A duplication check has just been done for your manuscript, and we found some overlapping sentences with previous works, which is not allowed by us.
Answer: Thank you for your comment. To avoid the duplications, we revised figure legends and supplemented the treatment of BGE and ginsenosides in methods section. Please refer to L81-83, L95-97, L109-111, L122-124, L135-137, L149-151, L188-193, L296-297, and L342-344 in revised manuscript.
Reviewer #1: The topic of the work is not very original, but it is a continuation or rather an extension of research on the medicinal properties of black ginseng extracts by some of the authors (probably from the same, larger research team), the results of which were previously published (Manuscript ID ijms- 2520355). Each new approach to a specific topic allows you to expand your knowledge in a specific area. Sometimes it confirms and sometimes excludes available data. In this aspect, the results presented by the authors are worth publishing. For the methods, I have previously reviewed other works by these authors. I then submitted comments and now I see that the authors also took them into account in the current manuscript, which helped avoid repeating random mistakes. I have no comments on the consistency of the applications. In my opinion, they refer to the assumed goal and the results of the research conducted. Similarly, the discussion is coherent and logical. References are selected and cited correctly. Formatting appropriate to IJMS requirements has also been preserved.
Answer: Thank you for your review for our manuscript and considerable comment. We would like to investigate on various materials to determine not only anti-neuroinflammatory activity but also various functions.
We are thankful to you and reviewers who give the corrections and very valuable suggestions through the whole manuscript. I hope the improved version will be acceptable for publication in IJMS
Sincerely yours,
Dr. Dae Young Lee

Reviewer 2 Report
This study shows that BGE inhibited several inflammatory events in LPS-induced BV2 cells. The story is theoretical according to the results and the methods used seem appropriate. I have some minor concerns.
1. The legend of Fig.7 (L164-170) is wrong.
2. As shown in Fig.7, mRNAs of TLR4 and MyD88 were changed by LPS and BGE. Is the activity of TLR4/MyD88 pathway dependent on the protein contents of these factors?
3. The concentration of each ingredient should be expressed with the dimension of ug/mL or so instead of uM in Fig.9. The inhibitory effects of ginsenoside Rd seems more potent than the other ginsenosides but the content of Rd in BGE is less than the others by less than 1/10. I’m wondering which ingredients the inhibitory effects of BGE is depending on.
4. The chemical differences of each ginsenoside should be explained.
5. In L263 “treatment with” should be “treatment with BGE”.
6. The method to dissolve BGE and some ingredients (ginsenoside) used in biological tests should be drawn. For example, to make 200ug/mL higher concentrations should be necessary.
7. The word, “finterated” in L405 should be “filtrated”.
Author Response
Dear Editors and Reviewers,
We greatly appreciate your kind consideration on our manuscript. According to the Reviewers’ comments, some parts of the manuscript have been corrected and revised.
Editor comments: Please rewrite lines 81-86, 130-133, 147-151, 167-168, 320-325. A duplication check has just been done for your manuscript, and we found some overlapping sentences with previous works, which is not allowed by us.
Answer: Thank you for your comment. To avoid the duplications, we revised figure legends and supplemented the treatment of BGE and ginsenosides in methods section. Please refer to L81-83, L95-97, L109-111, L122-124, L135-137, L149-151, L188-193, L296-297, and L342-344 in revised manuscript.
Reviewer #2: This study shows that BGE inhibited several inflammatory events in LPS-induced BV2 cells. The story is theoretical according to the results and the methods used seem appropriate. I have some minor concerns.
- The legend of Fig.7 (L164-170) is wrong.
Answer: Thank you for your comment. We revised the legend of Fig. 7. Please refer to L149-151 in revised manuscript.
- As shown in Fig.7, mRNAs of TLR4 and MyD88 were changed by LPS and BGE. Is the activity of TLR4/MyD88 pathway dependent on the protein contents of these factors?
Answer: Thank you for your comment. It is known that the activity of NF-κB and MAPK signaling pathways and the expression of pro-inflammatory mediators reviewed in this investigation are mainly regulated by the TLR4/MyD88 dependent pathway [1]. However, these factors could also be controlled by other signaling pathway other than MyD88 including tumor necrosis factor receptor-associated factor (TRAF), toll/interleukin 1 (TIR) domain-containing adapter-inducing IFNβ (TRIF), and TRIF-related adapter molecule (TRAM) [2, 3]. Therefore, it is necessary to further examine the effect of black ginseng extract on downstream factors of TLR4 such as TRAF, TRIF, and TRAM, and we are willing to identify another molecular mechanism of anti-neuroinflammatory effect of black ginseng. We hope that this answer would correspond to your comment.
References
- Duan, T.; Du, Y.; Xing, C.; Wang, H.Y.; Wang, R.F. Toll-Like Receptor Signaling and Its Role in Cell-Mediated Immunity. Front Immunol. 2022, 13, 812774.
- Patel, H.; Shaw, S.G.; Shi-Wen, X.; Abraham, D.; Baker, D.M.; Tsui, J.C.S. Toll-like receptors in ischaemia and its potential role in the pathophysiology of muscle damage in critical limb ischaemia. Cardiol Res Pract. 2012, 2012, 121237.
- Zamyatina, A.; Heine, H. Lipopolysaccharide Recognition in the Crossroads of TLR4 and Caspase-4/11 Mediated Inflammatory Pathways. Front Immunol. 2020, 11, 585146.
- The concentration of each ingredient should be expressed with the dimension of ug/mL or so instead of uM in Fig.9. The inhibitory effects of ginsenoside Rd seems more potent than the other ginsenosides but the content of Rd in BGE is less than the others by less than 1/10. I’m wondering which ingredients the inhibitory effects of BGE is depending on.
Answer: Thank you for your comment. The experimental results in Fig. 9. are the results of comparing the levels of inhibitory activity of each ginsenoside contained in BGE without considering the content of ginsenosides contained in BGE. As your comment, the inhibitory effects of ginsenoside Rd showed more potent than other ginsenosides, and the content of Rd in BGE is less than the others by less than 1/10. Our results suggest that the inhibitory effect of BGE is mainly regulated by Rd and Rg3, but further experiments are considered necessary considering the level of ginsenoside content contained in BGE. We hope that this answer would correspond to your comment.
- The chemical differences of each ginsenoside should be explained.
Answer: : Thanks for your comments. As your pointed out, we have added the sentence in the manuscript as below.
Effect on the production of NO and pro-inflammatory cytokines induced by LPS of Rd was significantly higher than those of the other ginsenoside (Rg3, Rg5, Rk1) (Figure 9). The chemical structures of Rg3, Rg5, Rk1 and Rd are the similar but they have dif-ferent sugar groups. In particular, Rd has one more glucose group at C-21 than other ginsenosides. This result indicates that the glucose group at carbon # 21 of Rd is one of the key factors for the anti-inflammatory effects of this molecule.
- In L263 “treatment with” should be “treatment with BGE”.
Answer: Thank you for your comment. We revised the correspond sentence. Please refer to L240 in revised manuscript.
- The method to dissolve BGE and some ingredients (ginsenoside) used in biological tests should be drawn. For example, to make 200ug/mL higher concentrations should be necessary.
Answer: Thank you for your comment. We described methods about dissolution of BGE and ginsenosides and their dilution for cell treatment. Please refer to L319 and L323-324 in revised manuscript.
- The word, “finterated” in L405 should be “filtrated”.
Answer: Thank you for your comment. We corrected the corresponding word. Please refer to L380 in revised manuscript.
Additional modifications
Since the contents of four ingredients including ginsenoside Rd, Rg3, Rk1, and Rg5 were examined in Figure 8, the contents of Rg3-R were removed in Figure 9, which was also applied to part of the methods and discussion section. The corresponding modifications are indicated in blue color.
We are thankful to you and reviewers who give the corrections and very valuable suggestions through the whole manuscript. I hope the improved version will be acceptable for publication in IJMS
Sincerely yours,
Dr. Dae Young Lee
